# Cardiodynamic variables measured by impedance cardiography during a 6-minute walk test are reliable predictors of peak oxygen consumption in young healthy adults

Fang Liu[1]☯, Raymond C. C. Tsang[2]☯, Alice Y. M. Jones[3]☯*, Mingchao Zhou[1]‡, Kaiwen Xue[4]‡, Miaoling Chen[1]☯, Yulong Wang[1]☯*

**1** Department of Rehabilitation, Shenzhen Second People's Hospital, The First Affiliated Hospital of Shenzhen University Health Science Centre, Shenzhen, China, **2** Department of Physiotherapy, MacLehose Medical Rehabilitation Centre, Hong Kong, China, **3** School of Health and Rehabilitation Sciences, The University of Queensland, Queensland, Australia, **4** School of Rehabilitation Sciences, The Shandong University of Traditional Chinese Medicine, Shandong, China

☯ These authors contributed equally to this work.
‡ MZ, KX and MC also contributed equally to this work.
* a.jones15@uq.edu.au (AYMJ); ylwang668@163.com (YW)

**Data Availability Statement:** All relevant data are within the manuscript and its Supporting Information files.

## Abstract

Accurate prediction of aerobic capacity is necessary to guide appropriate exercise prescription. It is common to use 6-minute walk distance (6MWD) to predict peak oxygen uptake ($VO_{2peak}$) in the clinical environment. The aim of this study was to determine whether prediction of $VO_{2peak}$ can be improved by the inclusion of cardiovascular indices derived by impedance cardiography (ICG) during the 6MWT. A total of 62 healthy university students aged 21±1 years completed in separate days, a cardiopulmonary exercise test (CPET) and two 6MWTs (30 min apart), during which heart rate (HR), stroke volume (SV) and cardiac output (CO) were measured by ICG (PhysioFlow® PF07 Enduro™). The CPET was conducted with the Ergoselect 200 Ergoline and oxygen consumption measured by a MasterScreen™ CPX breath-by-breath metabolic cart. Multiple regression analyses were conducted to generate $VO_{2peak}$ prediction equations using 6MWD with, or without the cardiovascular indices recorded at the end of the best performed 6MWT as predictor variables. The mean peak HR (bpm), SV (ml) and CO (L/min) recorded during 6MWT were 156±18, 95.6±9, 15±2.8 and during CPET were 176±16, 91.3±8, 16.2±2.7, respectively. Analyses revealed the following $VO_{2peak}$ prediction equation: $VO_{2peak}$ = 100.297+(0.019x6MWD)+(-0.598xHR$_{6MWT}$)+ (-1.236xSV$_{6MWT}$) + (8.671 x CO$_{6MWT}$). This equation has a squared multiple correlation ($R^2$) of 0.866, standard error of the estimate (SEE) of 2.28 mL/kg/min and SEE:$VO_{2peak}$ (SEE%) of 7.2%. Cross-validation of equation stability using predicted residual sum of squares (PRESS) statistics showed a $R^2$ ($R_p^2$), SEE (SEE$_p$) and SEE$_p$% of 0.842, 2.38 mL/kg/min and 7.6% respectively. The minimal shrinkage of $R^2$ implied regression model stability. Correlation between measured and predicted $VO_{2peak}$ using this equation was strong (r = 0.931, p<0.001). When 6MWD alone was used as the predictor for $VO_{2peak}$, the generated equation had a lower $R^2$ (0.549), and a higher SEE (4.08 mL/kg/min) and SEE% (12.9%). This is

**Funding:** This research was funded by the Sanming Project of Medicine in Shenzhen. Grant number SZSM201512011.

**Competing interests:** The authors have declared that no competing interests exist.

the first study which included cardiac indices during a 6MWT as variables for $VO_{2peak}$ prediction. Our results suggest that inclusion of cardiac indices measured during the 6MWT more accurately predicts $VO_{2peak}$ than using 6MWD data alone.

## Introduction

Cardiorespiratory fitness is an essential component of health. The gold standard for evaluation of aerobic capacity is through a graded cardiopulmonary exercise test (CPET) [1]. However, a CPET requires expensive equipment and trained-personnel to administer the test [2]. A 6-minute walk test (6MWT) is a field test commonly used to assess functional capacity in people with chronic cardiopulmonary diseases [3–5]. The 6-minute walk distance (6MWD) is reportedly an accurate predictor of peak oxygen consumption ($VO_{2peak}$) in both paediatric [6] and older adult populations [7, 8]. One determinant of maximal oxygen consumption is cardiac output, which in turn is influenced by stroke volume and heart rate. While heart rate can easily be measured during a 6MWT, non-invasive measurements of CO and SV during a 6MWT have not been regularly considered. Impedance cardiography (ICG) is a non-invasive technique which provides essential indices of cardiovascular function, including HR, SV and CO. Our previous work showed that in a post-stroke population, ICG provides reliable, detailed, non-invasive cardiodynamic data during a 6MWT [9]. We hypothesise that inclusion of ICG measured data during a 6MWT will produce a stronger predictor of $VO_{2peak}$, compared to 6MWD alone. The aim of this study was to investigate whether the inclusion of cardiovascular indices recorded during a 6MWT improved the accuracy and stability of a regression model predicting $VO_{2peak}$, compared to a predictive equation using 6MWD alone.

## Materials and methods

This was a cross-section observational study approved by the Institutional Review Board of Shenzhen Second People's Hospital (Ethics approval number: 20191211001). The study protocol (Register number: ChiCTR2000028771) is available at the Chinese Clinical Trial Register Center website: http://www.chictr.org.cn/index.aspx. Protocol of this study is also available at Protocol.io, dx.doi.org/10.17504/protocols.io.bupdnvi6.

### Participants

University students with normal health were recruited to participate in this study. Invitation emails were sent via the student portal of all students registered at the School of Rehabilitation of a local university affiliated with the hospital where this project was conducted. Exclusion criteria were: (1) students with known cardiovascular, pulmonary or musculoskeletal disorders that might limit their participation in maximal ergometric exercise test; (2) pregnancy; and (3) any anxiety/depression psychiatric disorder.

### Procedures

Students who responded to the invitation email were invited to the involved hospital. The cardiopulmonary exercise test (CPET) and 6-minute walk test (6MWT) processes were explained, and signed informed consent was obtained. Testing was determined randomly by drawing lots from an envelope, whereby each participant on separate days completed, either the CPET, or, two consecutive 6MWT spaced 30 minutes apart, as the first test. Participants were invited to attend the cardiopulmonary laboratory of the hospital at a time >2 hours after a light meal. All

participants were requested to avoid caffeine-containing products, nicotine, and alcohol at least 12 hours before attending the laboratory.

Cardiodynamic parameters including HR, SV, CO were recorded at one-second intervals by means of ICG (PhysioFlow® PF07 Enduro™ Paris, France) during 6MWT and CPET. Oxygen saturation (SpO$_2$) was recorded with the Heal Force pulse oximeter (POD-3, China). Systolic and diastolic blood pressure (SBP and DBP) was measured with the OMRON electronic blood pressure monitor (U30, China). Rate of perceived exertion (RPE) at the end of each test was recorded with the modified 0–10 Borg Scale [10].

**6MWT.** The 6MWT was performed in a 30-meter hospital hallway, following the standard protocol recommended by the American Thoracic Society (ATS) [11]. This study design required comparison of cardiodynamic variables during a 6MWT and a CPET, thus two trials of 6MWT, with a rest period of at least 30 minutes between tests, were recorded to minimise the risk of possible learning effect and to ensure maximal effort. During data collection, participants were also asked to rest in a sitting position for 10 minutes before and after the 6MWT. The cardiodynamic parameters (HR, SV and CO) were measured using the ICG at one-second intervals during both tests. The SpO$_2$ was recorded before and after each 6MWT. The SBP and DBP were measured at two-minute intervals during the 10-minute rest period, before and after each 6MWT. The RPE was recorded immediately at the end of each 6MWT.

**CPET.** Each participant also performed a progressive CPET in the hospital cardiopulmonary laboratory using a cycle ergometer (Ergoselect 200, Ergoline GmbH, Germany). Throughout the test, 12-lead electrocardiography was continuously recorded, and the participant was required to wear a mask and breathe through a calibrated volume sensor. Oxygen consumption, carbon dioxide consumption and respiratory exchange ratio (RER) were measured by the MasterScreen™ CPX, breath-by-breath metabolic cart (CareFusion, Germany).

The gas analysis system was fully calibrated immediately before each test in accordance with the manufacturer's instructions. The HR, SV, and CO were also measured with ICG at one-second intervals during the CPET.

The Garatachea 3-stage exercise protocol was adopted [12]. The first stage commenced with participants resting for three minutes in an upright position on the cycle ergometer with hands rested on the handlebars. Stage 2 required pedalling with free load for three minutes at a speed of 60 revolutions per minute. During Stage 3 the participant continued pedalling at the same speed but with a load of 25W, increased incrementally by 25W per minute, until the subject, despite encouragement, could no longer maintain the required speed, or, the participant's respiratory exchange quotient (REQ) exceeded 1.1 [12]. Participants were asked to rate their sensation of fatigue using the modified Borg scale. The cool down work rate was 15W at 60 revolutions per min for 3 minutes, followed by resting in a seated position.

**Impedance cardiography ICG.** The PhysioFlow® PF07 is a portable, non-invasive device that adopts real-time wireless monitoring of morphology-based impedance cardiography signals via a blue tooth USB adapter to measure HR, SV and CO at one-second intervals. Electrodes were applied as described by Tonelli and colleagues [13]. The HR was derived directly from the ECG [14], SV was calculated from the cardiac ejection waveform and the CO was obtained by the multiplication of the SV and HR [15]. ICG data during the last 10 seconds of the 6MWT and CPET were averaged as the peak cardiodynamic variables. This ICG measurement method was previously reported as reliable [9].

## Statistical analysis

All statistical analyses were performed using the IBM SPSS for Windows, version 25 (Armonk, NY: IBM Corp). Demographic data and clinical variables of all participants were summarized

using descriptive statistics. Variables of interval-ratio data meeting normality assumption were compared using independent t test for gender difference. Chi-square test was used for analysis of categorical data of different gender. The results from the 6MWT that produced a higher 6MWD were used for statistical analyses. A total of five multiple linear regression analyses were conducted. The first set of two analyses were performed to examine the unique contribution of HR or SV changes to the outcome variable 'change in CO', during the 6MWT and CPET. Forced entry regression method with HR change and SV change as predictor variables was used. The second set, which included three multiple linear regression analyses was conducted with $VO_{2peak}$ recorded during the CPET as the outcome variable to generate regression equations. In this set, the first regression equation was generated using age, gender, BMI, and $HR_{peak}$, $SV_{peak}$ and $CO_{peak}$ measured during CPET as predictor variables. The second regression equation was developed with variables of 6MWD, age, gender, BMI, plus ICG recorded HR, SV and CO at the end of the better performed 6MWT as predictors for $VO_{2peak}$. The third regression equation was created using 6MWD as the sole variable predicting $VO_{2peak}$. A stepwise backward regression method was then adopted to determine the statistically significant predictor variables retained in the regression equations. The appropriateness and precision of the regression parameters were evaluated with the squared multiple correlation ($R^2$), the standard error of estimate (SEE), and partial SEE (SEE %), which was 'SEE:mean$VO_{2peak}$' ratio expressed as a percentage. The predicted residual sum of squares (PRESS) statistic [16] was computed to estimate the degree of $R^2$ shrinkage when the $VO_{2peak}$ regression equation was used for cross-validation across similar but independent samples. PRESS-derived $R_p^2$, $SEE_p$ and $SEE_p$% were compared with those of the final regression models.

**Sample size estimation.** A sample size of at least 51 subjects was required for the multiple regression analyses using the PASS 15.0.5 (Kaysville, Utah: NCSS) for an estimated Cohen's effect size of 0.35 with 7 predictor variables, a level of significance of 0.05 and statistical power of 0.8 [17].

## Results

A total of 62 participants (25 males) with normal health aged 21.3±1.2 years old participated in this study between 20[th] December 2019 and 30[th] November 2020. All participants achieved a REQ of 1.1 within 12 minutes of commencing the CPET. The mean age, BMI, lean body mass, exercise habits and clinical characteristics are shown in Table 1.

The male participants were slightly older, had a higher BMI, lean body mass, 6MWD and achieved a higher peak work rate during CPET compared with the female participants. There were more female participants who exercised less than three times per week, but more male participants who exercised three to five times per week. Descriptive statistics of ICG measured HR, SV and CO for genders are displayed in Table 2.

The $VO_{2peak}$ achieved during CPET correlated strongly with peak ICG HR, SV and CO data measured during CPET and 6MWT (Table 3). However, the correlation between $VO_2$ recorded at anaerobic threshold (AT) during CPET and ICG cardiodynamic data recorded at the end of 6MWT were much lower (Table 3). Recorded $SpO_2$ and blood pressure responses were uneventful. (S1 Table).

### Regression analyses 1 and 2

Multiple linear regression analyses revealed that the changes in HR or SV contributed equally to the change of CO during the 6MWT. However, the contribution of HR changes was more than double the contribution of SV, to the change in CO during CPET (Table 4).

**Table 1. Demographic data of the 62 participants.** Data in mean±SD unless annotated otherwise.

| | Total cohort (n = 62) | Males (n = 25) | Females (n = 37) | P value[#] |
|---|---|---|---|---|
| Age (years) | 21.3±1.2 | 22.0±1.4 | 20.8±0.7 | <0.001 |
| BMI (kg/m2) | 21.4±3.0 | 22.9±3.3 | 20.4±2.3 | 0.001 |
| Lean body mass (kg) | 43.5±8.7 | 52.2±5.9 | 37.7±4.4 | <0.001 |
| Exercise habits | | | | |
| <3 times/week (%) | 50 (80.6%) | 16 (25.8%) | 34 (54.8%) | |
| 3–5 times/week (%) | 11 (17.7%) | 8 (12.9%) | 3 (4.8%) | <0.05 |
| >5 times/week (%) | 1 (1.6%) | 1 (1.6%) | 0 (0%) | |
| Age predicted HRmax | 199±1 | 198±1 | 199±1 | >0.05 |
| 6MWD–test 1 | 610±64 | 646±54 | 586±60 | <0.001 |
| 6MWD–test 2 | 630±75 | 670±64 | 603±70 | <0.001 |
| Borg score at the end of the test with greater 6MWD | 7±1 | 7±1 | 7±1 | >0.05 |
| Peak work rate achieved during CPET (W) | 136±41 | 177±29 | 108±19 | <0.001 |
| Borg score at the end of the CPET | 8±1 | 8±1 | 8±1 | >0.05 |

[#]Variables between gender were compared with independent t-test except "exercise habits", which were compared with Chi-square test.

HR$_{max}$ = maximal heart rate; 6MWD = distance covered in the 6-minute walk test; CPET = cardiopulmonary exercise test.

## Regression analysis 3

Regression analysis also revealed that SV$_{peak}$ and CO$_{peak}$ recorded at CPET were the final significant predictor variables for VO$_{2peak}$. This regression model: VO$_{2peak}$(mL/kg/min) = 29.335 + (-0.648 x SV$_{peak}$ (mL)) + (3.795 x CO$_{peak}$ (L/min)) was associated with a $R^2$ of 0.886, SEE of 2.07 mL/kg/min and SEE% of 6.6% (Model 1 in Table 5).

## Regression analysis 4

Applying multiple linear regression analyses using cardiodynamic data collected during the 6MWT as predictor variables and VO$_{2peak}$ recorded at CPET as the outcome variable derived the equation:

VO$_{2peak}$(mL/kg/min) = 100.297 + (0.019 x 6MWD(m)) + (-0.598 x HR$_{6MWT}$) + (-1.236 x SV$_{6MWT}$)+(8.671 x CO$_{6MWT}$).

The $R^2$ of this equation was 0.866, with SEE of 2.28 mL/kg/min, and SEE% of 7.2%. (Model 2A, Table 5). The PRESS derived regression model illustrated a $R_p^2$ of 0.842, SEE$_p$ of 2.38mL/kg/min and SEE$_p$% of 7.6%. The shrinkage of $R^2$ was only 2.4%.

In our subject cohort, correlation analysis between predicted VO$_{2peak}$ derived from equation Model 2A and the actual measured VO$_{2peak}$ during CPET showed that there was a strong correlation between the measured and predicted VO$_{2peak}$ (r = 0.931, p<0.001).

## Regression analysis 5

When 6MWD was used as the only predictor for VO$_{2peak}$, the equation generated was VO$_{2peak}$ = -19.248 + (0.076 x 6MWD(m)) (Model 2B, Table 5).

This equation had a lower $R^2$ of 0.549, and a higher SEE of 4.08 mL/kg/min and SEE% of 12.9%.

## Discussion

The 6MWD has been reported as a reliable predictor of VO$_{2peak}$ in both the paediatric [6] and older adult populations [7, 8], but this is the first study to show that the addition of

**Table 2. Cardiodynamic data (mean±SD) measured using ICG during 6MWT and CPET.**

| | 6MWT | | | | | | | Peak VO$_2$ (ml/min/kg) | CPET | | | | | |
|---|---|---|---|---|---|---|---|---|---|---|---|---|---|---|
| | 6MWD (m) | Heart rate | | Stroke volume | | Cardiac output | | | Heart rate | | Stroke volume | | Cardiac output | |
| | | RHR$_{6MWT}$ (bpm) | PHR$_{6MWT}$ (bpm) | RSV$_{6MWT}$ (ml) | PSV$_{6MWT}$ (ml) | RCO$_{6MWT}$ (L/min) | PCO$_{6MWT}$ (L/min) | | RHR$_{CPET}$ (bpm) | PHR$_{CPET}$ (bpm) | RSV$_{CPET}$ (ml) | PSV$_{CPET}$ (ml) | RCO$_{CPET}$ (L/min) | PCO$_{CPET}$ (L/min) |
| Male | 688±44 | 87±13 | 161±15 | 70.7±7.9 | 96.6±9.7 | 6.1±1.1 | 15.6±2.7 | 33.2±6.2 | 87±13 | 181±16 | 74.3±10.3 | 94.4±8.6 | 6.4±1.1 | 17.2±2.9 |
| Female | 660±65 | 85±10 | 152±19 | 66.5±7.7 | 95±8.5 | 5.6±0.9 | 14.6±2.8 | 30.4±5.7 | 86±11 | 173±15 | 67±8.1 | 89.1±7 | 5.8±1.1 | 15.5±2.3 |
| Total | 671±59 | 86±11 | 156±18 | 68.2±8 | 95.6±9 | 5.8±1 | 15±2.8 | 31.5±6 | 86±12 | 176±16 | 69.9±9.7 | 91.3±8 | 6±1.1 | 16.2±2.7 |

ICG = impedance cardiography; 6MWT = 6 minute walk test; CPET = cardiopulmonary exercise test; VO$_2$ = oxygen consumption; RHR$_{6MWT}$ = resting heart rate in 6MWT; PHR$_{6MWT}$ = peak heart rate in 6MWT; RSV$_{6MWT}$ = resting stroke volume in 6MWT; PSV$_{6MWT}$ = peak stroke volume in 6MWT; RCO$_{6MWT}$ = resting cardiac output in 6MWT; PCO$_{6MWT}$ = peak cardiac output in 6MWT test; RHR$_{CPET}$ = resting heart rate in CPET; PHR$_{CPET}$ = peak heart rate in CPET; RSV$_{CPET}$ = resting stroke volume in CPET; PSV$_{CPET}$ = peak stroke volume in CPET; RCO$_{CPET}$ = resting cardiac output in CPET; PCO$_{CPET}$ = peak cardiac output in CPET.

**Table 3. Correlations of VO$_{2peak}$ with peak HR, SV and CO measured by ICG during 6MWT and CPET and correlations of VO$_{2AT}$ with peak HR, SV and CO measured at the end of 6MWT.**

| Variable | Correlation with VO$_{2peak}$ | P value |
|---|---|---|
| | Pearson r | |
| HR$_{Peak}$ (bpm) | 0.93 | <0.001 |
| SV$_{Peak}$ (ml) | 0.70 | <0.001 |
| CO$_{Peak}$ (L/min) | 0.89 | <0.001 |
| HR$_{6MWT}$ (bpm) | 0.88 | <0.001 |
| SV$_{6MWT}$ (ml) | 0.75 | <0.001 |
| CO$_{6MWT}$ (L/min) | 0.90 | <0.001 |
| | Correlation with VO$_{2AT}$ | |
| | Pearson r | |
| HR$_{6MWT}$ (bpm) | 0.49 | <0.001 |
| SV$_{6MWT}$ (ml) | 0.35 | <0.05 |
| CO$_{6MWT}$ (L/min) | 0.46 | <0.001 |

6MWT = 6-minute walk test; CPET = cardiopulmonary exercise testing; VO$_{2peak}$ = peak oxygen consumption; HR$_{Peak}$ = peak heart rate at CPET; SV$_{Peak}$ = peak stroke volume at CPET; CO$_{Peak}$ = peak cardiac output at CPET; HR$_{6MWT}$ = heart rate at the end of 6MWT; SV$_{6MWT}$ = stroke volume at the end of 6MWT; CO$_{6MWT}$ = cardiac output at the end of 6MWT, AT = anaerobic threshold.

cardiodynamic variables recorded by ICG provides a more accurate prediction of VO$_{2peak}$ in a young healthy adult population. Although the maximal exertion level (RPE) differed between CPET (8/10) and 6MWT (7/10), measured VO$_{2peak}$ correlated strongly with the cardiodynamic variables recorded during both the 6MWT and the CPET (Table 3).

HR, SV and CO data measured non-invasively during an exercise test allows a comprehensive investigation of the cardiac responses to exercise. These cardiac parameters are relatively easily obtained by ICG technology, and recent advances in clinical applications of this technique are well described [18]. The current study showed that an increase in cardiac output during a 6MWT in healthy adults was contributed almost equally by changes in HR (28%) or

**Table 4. Contributions of unique changes in HR and SV to changes in CO (multiple linear regression analyses using change in CO as an outcome variable).**

| | At the end of 6MWT | | | CPET | | | | | |
|---|---|---|---|---|---|---|---|---|---|
| | | | | At anaerobic threshold | | | At the end of CPET | | |
| | male | female | all | male | female | all | male | female | all |
| HR change standardized beta coefficient | 0.59 | 0.59 | 0.61 | 0.82 | 0.64 | 0.71 | 0.75 | 0.64 | 0.72 |
| SV change standardized beta coefficient | 0.58 | 0.50 | 0.51 | 0.48 | 0.45 | 0.44 | 0.55 | 0.45 | 0.45 |
| HR change semipartial correlation | 0.56 | 0.46 | 0.53 | 0.82 | 0.50 | 0.64 | 0.75 | 0.54 | 0.68 |
| SV change semipartial correlation | 0.55 | 0.39 | 0.44 | 0.48 | 0.35 | 0.40 | 0.55 | 0.38 | 0.43 |
| HR change unique contribution to CO change (%) | 31 | 21 | 28 | 68 | 25 | 40 | 57 | 29 | 46 |
| SV change unique contribution to CO change (%) | 30 | 15 | 20 | 23 | 12 | 16 | 30 | 14 | 18 |

6MWT = 6-minute walk test; CPET = cardiopulmonary exercise testing; HR = heart rate; SV = stroke volume; CO = cardiac output.

**Table 5. Multiple regression analyses for prediction of VO$_{2peak}$.**

| Prediction Models | Coefficients | β | SEE (mL/kg/min) | SEE% | $R^2$ | SEEp (mL/kg/min) | SEEp% | $Rp^2$ |
|---|---|---|---|---|---|---|---|---|
| Model 1# | | | 2.07 | 6.6% | 0.886 | 2.14 | 6.8% | 0.871 |
| constant | 29.335 | | | | | | | |
| SV$_{peak}$ | -0.648 | -0.864* | | | | | | |
| CO$_{peak}$ | 3.795 | 1.688* | | | | | | |
| Model 2A# | | | 2.28 | 7.2% | 0.866 | 2.38 | 7.6% | 0.842 |
| constant | 100.297 | 0.182* | | | | | | |
| 6MWD | 0.019 | 1.788* | | | | | | |
| HR$_{6MWT}$ | -0.598 | 1.840* | | | | | | |
| SV$_{6MWT}$ | -1.236 | 4.077* | | | | | | |
| CO$_{6MWT}$ | 8.671 | 0.182* | | | | | | |
| Model 2B# | | | 4.08 | 12.9% | 0.549 | 4.12 | 13.1% | 0.523 |
| constant | -19.248 | | | | | | | |
| 6MWD | 0.076 | 0.741* | | | | | | |

#Model 1- 6MWD, SV$_{peak}$ and CO$_{peak}$ during CPET as predictor variables.

#Model 2A- 6MWD, HR$_{6MWT}$, SV$_{6MWT}$, CO$_{6MWT}$, as predictor variables.

#Model 2B- 6MWD alone as the predictor variable.

$R^2$ = squared multiple correlation; $Rp^2$ = predicted residual sum of squares (PRESS) derived squared multiple correlation; SEE% = partial SEE (SEE/mean of VO$_{2peak}$ × 100%); SEEp = PRESS derived standard error of estimate; SEEp% = PRESS derived partial SEE (SEEp/mean of VO$_{2peak}$ × 100%); β = standardized regression weights.

*P < .005.

SV (20%); however, the increase in CO during CPET appeared to be dominated by an increase in HR (46%) more than SV (18%) (Table 4). This is not surprising. As reflected by a higher RPE, CPET demanded a higher workload than the 6MWT; at a high workload, increases in heart rate limit diastolic filling time, leading to a decrease in end-diastolic volume, which subsequently limits the increase in SV [19]. This accords with the higher maximal HR recorded at the end of a CPET compared to that recorded at the end of a 6MWT, while the maximal SV achieved was similar (Table 2). Interestingly, Table 4 also shows that the increased contribution to CO by HR, rather than SV, as a response to exercise, began at AT. Be that as it may, although the peak HR, SV and CO achieved at the end of 6MWT were not as high as the data measured at the end of a CPET, the correlation between oxygen consumption and cardiodynamic parameters measured at the end of 6MWT was strongest with VO$_2$ peak and not with oxygen consumption at AT (Table 3). This finding suggests that although submaximal, 6MWT induced exercise responses beyond AT and closer to peak CPET levels.

The gold standard for evaluation of aerobic capacity uses CPET to measure VO$_{2peak}$ (maximal oxygen consumption) [1]. Various predictors for VO$_{2peak}$ during a CPET have been reported and include work rate, body weight, age and gender [20–22]. Our data suggested that the significant predictors for VO$_{2peak}$ in our subject cohort were ICG-derived SV$_{peak}$ and CO$_{peak}$ during the CPET ($R^2$ 0.89). Clinically, CPET is not always practical, and submaximal field tests such as 6MWT are therefore relied upon to provide a convenient indication of aerobic capacity. Cardiodynamic parameters induced by a 6MWT were strongly correlated to similar cardiac parameter responses induced by a CPET, as seen in Table 3. Our study showed that the inclusion of cardiodynamic data recorded by ICG during the 6MWT accurately predict the VO$_{2peak}$ in our subject cohort. The $R^2$ value for this regression model was 0.87, which is comparable to the CPET regression model (0.89). Furthermore, PRESS analysis shows a $R^2$

shrinkage of only 2.4%, suggesting that inclusion of the cardiac predictor variables further stabilised the prediction equation.

When 6MWD alone was used as the predictor variable for VO$_{2peak}$, the $R^2$ was only 0.55, and SEE% almost doubled (13%). The use of 6MWD as a predictor variable for VO$_{2peak}$ has been reported in the literature [6–8] (Table 6). The $R^2$ and SEE values generated in different regression equations varied. Direct comparison between our regression models and others is not deemed appropriate, because the predictor variables reported were different. Age, gender and BMI have been used by others [6–8]; however, these variables were not significant predictors in our final regression equation. Exclusion of these variables in the VO$_{2peak}$ prediction equation was also reported in Sperandio et al study [8]. Furthermore, apart from different predictor variables entered for regression analysis, variation in age range and ethnicity of the sample population may also limit a meaningful comparison between regression equations [6].

## Implications of the study

ICG is a technique which is non-invasive and simple to employ. It also provides appropriate data for meaningful analysis of the cardiac responses to activity or exercise. Exercise programs are often prescribed by heath-professionals in community health centres. Prediction of

**Table 6. VO$_{2peak}$ prediction equations using 6-minute walk distance (6MWD) as predictor variable.**

| Studies | Age (year) | Gender (n) | 6MWD (m) | VO2peak (mL/kg/min) | Equation | $R^2$ | SEE (mL/kg/min) | SEE% |
|---|---|---|---|---|---|---|---|---|
| Current study | 21.3 ± 1.2 (19–26) | Males (25) | Males: 688 ± 44 | Males: 33.2 ± 6.2 | #Model 2A: | 0.866 | 2.28 | 7.2% |
| | | Females (37) | Females: 660 ± 65 | Females: 30.4 ± 5.7 | #Model 2B: | 0.549 | 4.08 | 12.9% |
| | | | Total: 671 ± 59 | Total: 31.5 ± 6.0 | | | | |
| Sperandio et al 2015 [8] | 54 ± 10 (40–74) | Males (40) | 608 ± 97 | 35 ± 11 | #Sperandio equation | 0.76 | - | - |
| | | Females (46) | | | | | | |
| Jalili et al 2018 [6] | 12.49±2.72 (8–17) | Boys (349) | 716 ± 60 | 41.24 ± 6.00 | #Jalili equation | 0.79 | 2.91 | 6.9% |
| Mänttäri et al 2018 [7] | 50±13 (19–75) | Males (39) | Males: 660 ± 84 | Males: 35.2 ±7.8 | #Mänttäri @Males: | 0.82 | 3.6 | 9.5% |
| | | Females (36) | Females: 624±62 | Females: 33.6 ±7.3 | #Mänttäri @Females: | 0.89 | 3.5 | 9.3% |
| | | | Total: 652 ±74 | Total: 34.4 ±7.6 | | | | |

6MWD = 6-minute walk distance; BMI = body mass index; VO2peak/VO2max = peak oxygen consumption.

Equations

# Model 2A: VO$_{2peak}$ (mL/kg/min) = 100.297 + (0.019 x 6MWD(m)) + (-0.598x HR$_{6MWT}$ (bpm)) + (-1.236x SV$_{6MWT}$ (mL)) + (8.671 x CO$_{6MWT}$(L/min)) VO$_{2peak}$ (mL/kg/min) = -2.863 + (0.0563 x 6MWD(m)).

# Model 2B: VO$_{2peak}$ (mL/kg/min) = -19.248 + (0.076 x 6MWD(m)).

# Sperandio et al [8]: VO$_{2peak}$ (mL/kg/min) = -2.863 + (0.0563 x 6MWD(m)).

# Jalili et al [6]: VO$_{2max}$ (mL/kg/min) = 12.701 + (0.06 × 6MWD (m))–(0.732 × BMI (kg/m$^2$)).

# Mänttäri et al [7]

@ Males: VO$_{2max}$ (mL/kg/min) = 110.546 + 0.063 x 6MWD(m) - 0.250 x age—0.486 x BMI(kg/m$^2$) -0.420 x height (cm) - 0.109 x HR.

@ Females: VO$_{2max}$ (mL/kg/min) = 22.506–0.271 x weight(kg) + 0.051x 6MWD(m) - 0.065 x age.

exercise capacity should be used to guide appropriate exercise prescription. The findings of this study show a need for further investigation of appropriate VO$_{2peak}$ prediction equations in different disease population, using cardiodynamic parameters recorded by ICG during a 6MWT. Further, the potential benefit of using ICG to investigate the cardiodynamic response to exercise and specific intervention, before and after operation in various patients, warrants further investigation.

## Limitations of the study

The implications of our study are restricted to healthy young adults. In consequence the prediction equation cannot be generalised to population of older adults nor people with chronic illness. However, this study affirmed the hypothesis that cardiodynamic parameters recorded during the 6MWT are better predictors of aerobic capacity compared to the 6MWD alone. Application of portable telemetric measurement of oxygen consumption during the 6MWT would allow direct comparison of oxygen consumption recorded during 6MWT and CPET, however, telemetric equipment was not available in our laboratory. Such comparisons but without cardiodynamic data have been reported previously [7]. Further studies in populations with various disease backgrounds are necessary for meaningful application in prediction of aerobic capacity in different disease populations.

## Conclusions

The findings of this study suggest that a regression model in healthy young adults which includes as predictor variables HR, SV and CO measured by ICG, together with the 6MWD, is a better predictor of aerobic capacity than a regression model using 6MWD alone. Further studies are required to test this regression model in various patient cohorts.

## Supporting information

**S1 Table. Oxygen saturation and blood pressures at baseline, immediately post and 10 min post-CPET and post-6MWT.** Data are mean±SD. 6MWT = 6 minute walk test; CPET = cardiopulmonary exercise test; SpO2 = oxygen saturation measured by oximeter; SBP = systolic blood pressure; DBP = diastolic blood pressure. (DOCX)

## Author Contributions

**Conceptualization:** Fang Liu, Raymond C. C. Tsang, Alice Y. M. Jones, Yulong Wang.

**Data curation:** Fang Liu, Mingchao Zhou, Miaoling Chen.

**Formal analysis:** Fang Liu, Raymond C. C. Tsang, Alice Y. M. Jones.

**Funding acquisition:** Yulong Wang.

**Investigation:** Fang Liu, Raymond C. C. Tsang, Alice Y. M. Jones.

**Methodology:** Fang Liu, Raymond C. C. Tsang, Alice Y. M. Jones.

**Project administration:** Fang Liu, Mingchao Zhou, Kaiwen Xue, Miaoling Chen, Yulong Wang.

**Resources:** Fang Liu, Kaiwen Xue, Yulong Wang.

**Supervision:** Fang Liu.

**Validation:** Fang Liu, Raymond C. C. Tsang, Alice Y. M. Jones, Kaiwen Xue.

**Visualization:** Fang Liu, Raymond C. C. Tsang, Alice Y. M. Jones, Yulong Wang.

**Writing – original draft:** Fang Liu, Raymond C. C. Tsang, Alice Y. M. Jones.

**Writing – review & editing:** Fang Liu, Raymond C. C. Tsang, Alice Y. M. Jones.

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
