## [Decision Letter · Decision Letter 0]

27 Apr 2021

PONE-D-21-09792

Cardiodynamic variables measured by Impedance Cardiography during a 6-minute walk test are reliable predictors of peak oxygen consumption in young healthy adults

PLOS ONE

Dear Dr. Jones,

Thank you for submitting your manuscript to PLOS ONE. After careful consideration, we feel that it has merit but does not fully meet PLOS ONE’s publication criteria as it currently stands. Therefore, we invite you to submit a revised version of the manuscript that addresses the points raised during the review process.

For this initial cycle of the peer review process, it has been fortunate that three experienced reviewers have been able to contribute their time and expertise to considering this submission. On the whole their comments are very supportive of the project to date, with to follow some recommendations to further evolve the reporting of the work to enhance its reporting to the readership.

We look forward to receiving your revised manuscript.

Kind regards,

Shane Patman, PhD

Academic Editor

PLOS ONE

Journal Requirements:

'This research was funded by the Sanming Project of Medicine in Shenzhen. Grant number SZSM201512011.'

'The funders had no role in study design, data collection and analysis, decision to publish, or preparation of the manuscript.'

Reviewers' comments:

Reviewer's Responses to Questions

**Comments to the Author**

1. Is the manuscript technically sound, and do the data support the conclusions?

Reviewer #1: Yes

Reviewer #2: Yes

Reviewer #3: Yes

2. Has the statistical analysis been performed appropriately and rigorously? 

Reviewer #1: Yes

Reviewer #2: Yes

Reviewer #3: I Don't Know

3. Have the authors made all data underlying the findings in their manuscript fully available?

Reviewer #1: Yes

Reviewer #2: Yes

Reviewer #3: Yes

4. Is the manuscript presented in an intelligible fashion and written in standard English?

Reviewer #1: Yes

Reviewer #2: Yes

Reviewer #3: Yes

5. Review Comments to the Author

Reviewer #1: The manuscript "Cardiodynamic variables measured by Impedance Cardiography during a 6-minute

walk test are reliable predictors of peak oxygen consumption in young healthy adults" is very interesting and of high clinical utility. The design of the study is correct and thoughful, the aim is clear, the results support the conclusions. The disucussion is consisent and balanced. The proposed 6MWT/ICG and elaborated equation for calculation of VO2peak could be used in practice, i.e. for screening exams in young subjects. At first glance simply study effected in really valuable conclusions.

I have only minor comments to be cosidered:

1. Introduction, verse 38 "....measured data during a 6MWT will be a stronger predictor of...." - maybe better "....measured data during a 6MWT will produce a stronger predictor of...."

2. Methods, verse 68-70 - the sentence about MasterScreen CPX - please delete here and insert in sentence verse 85-86 in CPET description.

Reviewer #2: The present study investigates the use of impedance cardiography during a 6MWT for better predictability of VO2max. In general, the study seems well conducted and the manuscript is clear and easy to read. The authors should be complemented for addressing an important issue in the assessment of physical fitness in the clinical setting. I have only few comments for consideration.

Abstract:

- Line 10-21: A presentation of the peak values HR, SV, CO in CEPT and 6MWT would contribute to a better understanding of the study in the abstract. The description of the regression analyses could be more concise.

Introduction:

- Line 32-34: A little more detail is needed regarding the prediction of oxygen consumption due to the 6MWT.

Methods:

- Line 73: Why were two consecutive 6MWT conducted?

Discussion:

- Line 318: Measuring VO2 with a portable spirometry during the 6MWT would also be useful.

- Also, the feasibility nature of the study should be addressed.

The authors have demonstrated the benefits of impedance cardiography in simple clinical examinations. Further investigations (e.g. before and after operations) in various patient cohorts would be useful in the sequel. I recommend considering the manuscript for publication.

Reviewer #3: In the manuscript "Cardiodynamic variables measured by Impedance Cardiography during a 6-minute walk test are reliable predictors of peak oxygen consumption in young healthy adults" authors determine whether prediction of VO2peak can be improved by the inclusion of cardiovascular indices derived by impedance cardiography (ICG) during the 6MWT.

This manuscript is correctly written, with the connection between aim, results, discussion and conclusions preserved. In my opinion, it is of significant clinical importance, although it is very limited to the studied age group, without chronic diseases, according to the authors' opinion.

Minor comments:

1. While planning training or rehabilitation, it is important to determine the anaerobic threshold. Do you have any data on the level of effort achieved during the 6MWT (line 273-submaximal)? At the end of 6MWT, does the patient get VO2 corresponding to peak VO2 or VO2 at anaerobic threshold (AT) during CPET? Hence, there may be a difference in the strength of the correlation between hemodynamic parameters and VO2 between CPET and 6MWT. The incresases in SV and HR from their values at rest to those at the AT (AT–rest) are similar, but the increase in SV is significantly lower than HR after AT. Moreover, the increase in SV after AT is sensitive to lack of training and chronic diseases. Please complete the discussion on line 260-269.

2. How did you choose values for hemodynamic parameters suitable for analysis? How did you deal with measurement errors? Please complete the methodology - line 103.

6. PLOS authors have the option to publish the peer review history of their article (what does this mean?). If published, this will include your full peer review and any attached files.

Reviewer #1: No

Reviewer #2: No

Reviewer #3: **Yes: **Małgorzata Kurpaska, MD, PhD

---

## [Author Response · Author response to Decision Letter 0]

2 May 2021

We are grateful for all our reviewers’ valuable and insightful comments and have modified our manuscript following their suggestions.

Reviewer #1: 

The manuscript "Cardiodynamic variables measured by Impedance Cardiography during a 6-minute walk test are reliable predictors of peak oxygen consumption in young healthy adults" is very interesting and of high clinical utility. The design of the s

tudy is correct and thoughtful, the aim is clear, the results support the conclusions. The discussion is consistent and balanced. The proposed 6MWT/ICG and elaborated equation for calculation of VO2peak could be used in practice, i.e. for screening exams in young subjects. At first glance simply study effected in really valuable conclusions.

I have only minor comments to be considered:

1. Introduction, verse 38 "....measured data during a 6MWT will be a stronger predictor of...." - maybe better "....measured data during a 6MWT will produce a stronger predictor of...."

2. Methods, verse 68-70 - the sentence about MasterScreen CPX - please delete here and insert in sentence verse 85-86 in CPET description.

Authors responses:

We are grateful for Reviewer 1’s supportive comments. We have replaced ‘be’ with ‘produce’ in the sentence as advised. Please see line 43 under Introduction. 

We have now moved the description of MasterScreen CPX to the section under CPET, please see line 93.

Reviewer #2: 

The present study investigates the use of impedance cardiography during a 6MWT for better predictability of VO2max. In general, the study seems well conducted and the manuscript is clear and easy to read. The authors should be complemented for addressing an important issue in the assessment of physical fitness in the clinical setting. I have only few comments for consideration.

Authors response: We are grateful for Reviewer 2’s encouraging comments

Abstract:

- Line 10-21: A presentation of the peak values HR, SV, CO in CEPT and 6MWT would contribute to a better understanding of the study in the abstract. The description of the regression analyses could be more concise.

Authors response: The peak values of HR, SV, CO in CPET and 6MWT are now included in the Abstract. Units in the equation were removed so the equation now appears more concise.

Introduction:

- Lines 32-34: A little more detail is needed regarding the prediction of oxygen consumption due to the 6MWT.

Authors response: An extra sentence is now included in the Introduction to highlight the relationship between oxygen consumption and ICG cardiodynamic data during 6MWT. Please see lines 36-39.

Methods:

- Line 73: Why were two consecutive 6MWT conducted?

Authors response: the reason for conducting two 6MWTs is now included, please refer to lines 78-81.

Discussion:

- Line 318: Measuring VO2 with a portable spirometry during the 6MWT would also be useful.

Authors response: This is an excellent point. We were aware of the advantage of using a portable telemetric device to measure oxygen consumption during 6MWT. Unfortunately, we did not have the resources to acquire this equipment. This has now been addressed as a limitation. Please see lines 328-331.

Also, the feasibility nature of the study should be addressed. The authors have demonstrated the benefits of impedance cardiography in simple clinical examinations. Further investigations (e.g. before and after operations) in various patient cohorts would be useful in the sequel. I recommend considering the manuscript for publication.

Authors response: we have now included the feasibility point in the section “Implications of our study’, please refer to lines 319-321.

Reviewer #3: 

In the manuscript "Cardiodynamic variables measured by Impedance Cardiography during a 6-minute walk test are reliable predictors of peak oxygen consumption in young healthy adults" authors determine whether prediction of VO2peak can be improved by the inclusion of cardiovascular indices derived by impedance cardiography (ICG) during the 6MWT.

This manuscript is correctly written, with the connection between aim, results, discussion and conclusions preserved. In my opinion, it is of significant clinical importance, although it is very limited to the studied age group, without chronic diseases, according to the authors' opinion.

Authors response: We are thankful for Reviewer 3’s positive comments. We are currently extending our study to include subjects after stroke. 

Minor comments:

1. While planning training or rehabilitation, it is important to determine the anaerobic threshold. Do you have any data on the level of effort achieved during the 6MWT (line 273-submaximal)? At the end of 6MWT, does the patient get VO2 corresponding to peak VO2 or VO2 at anaerobic threshold (AT) during CPET? Hence, there may be a difference in the strength of the correlation between hemodynamic parameters and VO2 between CPET and 6MWT. The incresases in SV and HR from their values at rest to those at the AT (AT–rest) are similar, but the increase in SV is significantly lower than HR after AT. Moreover, the increase in SV after AT is sensitive to lack of training and chronic diseases. Please complete the discussion on line 260-269.

Authors response: Thank you for this excellent comment. Indeed we have collected data on VO2 at anaerobic threshold during CPET. We have reported that the RPE level achieved at the end of CPET was 8/10 and that for 6MWT was 7/10 (line 250) but we did not report our AT data because the correlation between cardiodynamic and VO2peak data was much stronger than correlation with VO2 at AT. As suggested by the reviewer, we now consider that this information would be a nice addition to our report. We have therefore included data recorded at AT in Tables 3 and 4, and in the Results section: lines 182-184, and the Discussion at lines 263-269 and lines 276-277.

2. How did you choose values for hemodynamic parameters suitable for analysis? How did you deal with measurement errors? Please complete the methodology - line 103.

Authors response: Thank you for pointing this out. We averaged the final 10 seconds of the ICG data recorded to provide the peak cardiodynamic variables of the 6MWT, and data for CPET were computed similarly. This measurement method was considered reliable as previously reported [9]. We have now included this in the Methodology section, lines 112-114.

---

## [Decision Letter · Decision Letter 1]

12 May 2021

Cardiodynamic variables measured by Impedance Cardiography during a 6-minute walk test are reliable predictors of peak oxygen consumption in young healthy adults

PONE-D-21-09792R1

Dear Dr. Jones,

We’re pleased to inform you that your manuscript has been judged scientifically suitable for publication and will be formally accepted for publication once it meets all outstanding technical requirements.

Kind regards,

Shane Patman, PhD

Academic Editor

PLOS ONE

Additional Editor Comments (optional):

Reviewers' comments:

Reviewer's Responses to Questions

**Comments to the Author**

1. If the authors have adequately addressed your comments raised in a previous round of review and you feel that this manuscript is now acceptable for publication, you may indicate that here to bypass the “Comments to the Author” section, enter your conflict of interest statement in the “Confidential to Editor” section, and submit your "Accept" recommendation.

Reviewer #2: All comments have been addressed

2. Is the manuscript technically sound, and do the data support the conclusions?

Reviewer #2: Yes

3. Has the statistical analysis been performed appropriately and rigorously? 

Reviewer #2: Yes

4. Have the authors made all data underlying the findings in their manuscript fully available?

Reviewer #2: (No Response)

5. Is the manuscript presented in an intelligible fashion and written in standard English?

Reviewer #2: Yes

6. Review Comments to the Author

Reviewer #2: The authors have made the adjustments requested and I am happy that these changes have improved the article.

7. PLOS authors have the option to publish the peer review history of their article (what does this mean?). If published, this will include your full peer review and any attached files.

Reviewer #2: No

---

## [Editor Report · Acceptance letter]

17 May 2021

PONE-D-21-09792R1 

Cardiodynamic variables measured by Impedance Cardiography during a 6-minute walk test are reliable predictors of peak oxygen consumption in young healthy adults 

Dear Dr. Jones:

I'm pleased to inform you that your manuscript has been deemed suitable for publication in PLOS ONE. Congratulations! Your manuscript is now with our production department. 

Kind regards, 

on behalf of

Assoc Prof Shane Patman 

Academic Editor

PLOS ONE